# Snow Cover Response to Climatological Factors at the Beas River Basin of W. Himalayas from MODIS and ERA5 Datasets

**DOI:** 10.3390/s23208387

**Published:** 2023-10-11

**Authors:** Pardeep Kumar Gupta, George P. Petropoulos, Hemendra Singh Gusain, Vishakha Sood, Dileep Kumar Gupta, Sartajvir Singh, Abhay Kumar Singh

**Affiliations:** 1Department of Civil Engineering, Punjab Engineering College (Deemed to be University), Chandigarh 160012, India; sunita.ce@pbi.ac.in (S.); pardeepgupta@pec.edu.in (P.K.G.); 2Department of Civil Engineering, Punjabi University, Patiala 147002, India; 3Department of Geography, Harokopio University of Athens, El. Venizelou St., 70, Kallithea, 17671 Athens, Greece; gpetropoulos@hua.gr; 4Institute of Technology Management (DRDO), Mussoorie 248179, India; hs.gusain.dgre@gov.in; 5Department of Civil Engineering, Indian Institute of Technology, Ropar 140001, India; vishakha.sood@ieee.org; 6Department of Physics, Banaras Hindu University, Varanasi 221005, India; singhak@bhu.ac.in; 7University Institute of Engineering, Chandigarh University, Mohali 140413, India; sartajvir.dhillon@ieee.org

**Keywords:** snow cover area (SCA), climatological factors, Himalayas, permafrost, European Centre for Medium-Range Weather Forecasts (ECMWF)

## Abstract

Glaciers and snow are critical components of the hydrological cycle in the Himalayan region, and they play a vital role in river runoff. Therefore, it is crucial to monitor the glaciers and snow cover on a spatiotemporal basis to better understand the changes in their dynamics and their impact on river runoff. A significant amount of data is necessary to comprehend the dynamics of snow. Yet, the absence of weather stations in inaccessible locations and high elevation present multiple challenges for researchers through field surveys. However, the advancements made in remote sensing have become an effective tool for studying snow. In this article, the snow cover area (SCA) was analysed over the Beas River basin, Western Himalayas for the period 2003 to 2018. Moreover, its sensitivity towards temperature and precipitation was also analysed. To perform the analysis, two datasets, i.e., MODIS-based MOYDGL06 products for SCA estimation and the European Centre for Medium-Range Weather Forecasts (ECMWF) Atmospheric Reanalysis of the Global Climate (ERA5) for climate data were utilized. Results showed an average SCA of ~56% of its total area, with the highest annual SCA recorded in 2014 at ~61.84%. Conversely, the lowest annual SCA occurred in 2016, reaching ~49.2%. Notably, fluctuations in SCA are highly influenced by temperature, as evidenced by the strong connection between annual and seasonal SCA and temperature. The present study findings can have significant applications in fields such as water resource management, climate studies, and disaster management.

## 1. Introduction

The Himalayan region is known as the “Asian water tower” since it compromises the largest snow and ice cover outside the polar field [1]. Snow is a valuable asset in alpine regions, serving not only as a key attraction for tourism [2,3,4] but also playing a vital role in hydropower generation, water supply [5], and supporting the ecological aspects of local mountain flora and fauna [6]. Similarly, snow also holds significant importance in terms of natural hazards, including the prevention of avalanches and the forecasting of floods [7]. Moreover, climate fluctuations have a significant impact on the variability of snow cover [8]. The increasing global temperatures have led to the degradation of snow cover and the shrinking of glaciers, which in turn has contributed to increasing sea levels and alterations in the global climate [9,10,11], also significantly affecting permafrost regions. Snow cover changes both spatially and temporally throughout the multiple seasons of the year. Changes in the extent of SCAs, whether they decrease or increase over a certain period, serve as indicators of shifts in weather patterns. Long-term variations in snow and ice cover, spanning over three decades, indicate atmospheric climate change [12]. Consistent snow cover and glacier ice account for around 70% of the Earth system’s freshwater [13]. Snow cover in the Himalayan region in particular has a direct influence on the economy and society of neighboring countries such as North India. Knowledge about snow cover has been applied in forecasting runoff and snowmelt, as well as in the verification or adjustment of various hydrological models [14]. Moreover, the knowledge of snow cover and snowmelt runoff is also vital from the point of view of planning and operation of river valley projects and the perceptiveness of hydro-meteorological projects. It becomes very crucial to make a correct assessment of water volume in the basin gathered in the form of snow and when it melts due to temperature variations as its rate of release needs to be accurately predicted [15,16]. Both underestimation and overestimation can lead to harmful consequences such as floods, overflow of rivers, and shortage of water supply.

Various studies have indicated that the impacts of climate change are noticeable across the Himalayan cryosphere [17]. The current changes in the environment have caused excessive warming of certain regions, including permafrost areas such as those in the Arctic, which has led to the melting of glaciers [18,19]. This in turn has affected the glaciers and SCAs leading to their reduction [20,21]. As the temperature changes due to seasonal variations, it affects the snow cover and the extent of permafrost regions. Changes in SCA over a certain period can give an idea of the environmental changes or climate changes [22]. Snow cover affects the various scales and cycles directly or indirectly [23]. These include atmosphere, biological and ecosystem scales, and cycles [24]. So, it is very important to study the snow cover as it also affects the aquatic cycle management and the Earth’s hydrology, which further regulate alternations in climate and environmental change [25,26,27].

In the scientific literature, numerous approaches have been suggested for detecting changes in snow cover using normalized difference indices such as the normalized difference snow index (NDSI) [28,29], creating snow depth maps through field observations or surface temperature analysis [30,31], employing change detection methods, and utilizing pan-sharpening techniques [32]. The typical way of tracking snow cover entails gathering data on snow cover from weather stations or observation locations [33]. Nevertheless, this method faces challenges related to spatial consistency caused by the uneven distribution of these stations, which are predominantly located in low-altitude areas [34]. Accurately identifying alterations in snow coverage remains a challenging task due to multiple factors. For example, the rough geography of mountainous regions, the influence of shadow effects, and the ever-changing climatic conditions [35,36]. Change detection plays a crucial role in analysing temporal variations and visually representing diverse changes on a unified map [37]. It becomes very crucial to make a correct assessment of water volume in the basin gathered in the form of snow and when it melts due to temperature variations as its rate of release should be accurately predicted [38,39,40]. Both underestimation and overestimation can lead to harmful consequences like floods, overflow in rivers, or shortage of water supply, etc.

In a study by [41], the authors reported a positive trend in the SCA in the Indian Himalayas and the adjoining, Karakoram region, via the MODIS 8-day snow cover product for trend analysis. Similarly, [42] conducted a study on the variability of snow cover in the Kashmir Himalayas by analysing data from the 8-day MODIS snow cover product spanning from 2000 to 2016. Their findings revealed a consistent increase in SCA during this period, which aligns with the conclusions drawn by [43]. A study by [33] examined the variations in snow SCA within the climatic zones of the Indian Himalayas. The authors utilized MODIS sensor images to create a 10-day snow cover product covering the period from 2001 to 2016. Their findings indicated a change in the SCA trends over the Indian Himalayas and climatic zones after 2010. This suggests that the rate of change in snow cover may differ before and after this specific period, implying that climate change might be influencing the variability of snow cover in intricate ways. Due to the significant topographical variability and differences in study periods, there is a lack of consensus on the trends in SCA across different Himalayan regions. Additionally, there have been limited studies on SCA in the Indian Himalayan region, despite its significance. Cloud cover and sensor limitations may also constrain the accuracy of studies on SCA in the Himalayas. This is a common challenge when studying snow cover extent using remote sensing data, as clouds can obscure the ground and make it difficult to accurately assess snow cover [31]. Similarly, different sensors may have different limitations or capabilities when it comes to detecting snow cover, which can introduce additional sources of error.

Numerous research endeavours have been conducted in the Himalayan region, examining the variability of SCA. These studies have revealed trends, either increasing or decreasing, in response to the impacts of climate change. These studies encompass a range of terrain and meteorological factors, including factors such as elevation, temperature, aspect, and slope. Of all the parameters, elevation is the most important parameter in snow accumulation [43]. As the temperature can be inversely proportional to the elevation, there will be more snow accumulation at higher elevations, which ultimately leads to an increase in a snow-covered area [36]. As the temperature changes due to seasonal variations or for any other reason, it affects the snow cover. The alteration in temperature and precipitation patterns can significantly impact the extent and duration of snow cover, which in turn affects a range of ecological and social systems [35].

Herein, SCA dynamics were studied for the years from 2003 to 2018 in the Beas River basin of the Western Himalayas. SCA was estimated using the 8-day snow cover improved product (version 6) Combined Terra-Aqua (MOYDGL06*) derived from MODIS sensor images. The present study primarily focuses on analysing (a) the annual and seasonal variations in the SCA within the Beas River basin, (b) how SCA is influenced by factors such as elevation, aspect, and slope over the period from 2003 to 2018, and (c) to establish a relationship between temperature and precipitation by utilizing ERA5 data.

## 2. Materials and Methods

### 2.1. Study Area

The Beas River is an important river of the Indus River system, originating from the Western Himalayas (India), and its tributaries contribute to its flow (See Figure 1, AWiFS satellite image, downloaded from earthexplorer.usgs.gov, accessed on 9 June 2019). The basin’s size of 20,303 km^2^ indicates that it covers a large area, and the region’s altitude of 3900 m suggests that it is in the mountainous areas. The Beas Glacier near Rohtang Pass is the starting point of the river, and it flows almost north–south until the Larji Hydropower project, where it makes a right turn to the southwest and flows in the same direction until Pandoh Dam. The basin’s importance is evident from the fact that it provides water for irrigation, hydropower generation, and domestic use in the region. It is also a crucial part of the Indus River system, which is a significant river system in the Indian subcontinent.

The river’s length of 116 kilometers from its source to Pandoh Dam and the area of the catchment is 5384 km^2^. The glacier-fed Parbati and SainjKhad Rivers are essential tributaries of the Beas River, contributing to its flow. The Parbati River joining near Bhuntar, Sabari Nala near Kulu, Tirthan, and Sainj Rivers near Larji, and the Bakhli near Pandoh Dam are all significant tributaries that contribute to the Beas River flow. The Pandoh Hydropower Project, built to transport water from the Beas River to the Satluj River, is an essential structure that facilitates water management and power generation in the region. The Dehar Power House on the Satluj River right bank is an important power generation facility that uses this diverted stream from the Beas River to generate electricity. To establish a correlation between SCA and elevation, the study area was divided into seven elevation zones. A summary of the various elevation zones extent is shown in Table 1a, slope zones in Table 1b, and aspect zones in Table 1c.

### 2.2. Dataset

#### 2.2.1. Modis Product

This research utilized the MODIS product, i.e., MOYDGL06 maximum-snow-extent result of 8-day data from 2003–2018. Version 6 of MOYDGL allows the minimization of omission and commission errors, primarily in clear-sky situations. The implemented method incorporated the integration of the Terra-Aqua 8-day MODIS snow product version 6 (MOD10A2.006* and MYD10A2.006*) with the Randolph Glacier Inventory (RGI6.0). This integration process included several steps and filters to eliminate clouds, such as seasonal, temporal, and spatial filtering. The combined Terra and Aqua data, along with the glaciers from RGI 6.0, were used to generate the final snow product. A detailed explanation of this methodology can be found in [44]. Data were acquired from 1 January 2003 to 27 December 2018. A total of 736 MODIS images were used in the present study.

#### 2.2.2. ASTER DEM

Advanced Spaceborne Thermal Emission and Reflection Radiometer (ASTER) Digital Elevation Model Version (DEM) was obtained from (https://earthexplorer.usgs.gov/, accessed on 24 January 2021) to measure elevation, slope, and aspect parameters. The ASTER DEM has a resolution of 30 m. In this study, the DEM was resampled at 500 m resolution.

#### 2.2.3. Precipitation and Temperature Data

For analysis of the trend and its correlation analysis with SCA, data from the fifth-generation European Centre for Medium-Range Weather Forecasts (ECMWF) Atmospheric Reanalysis of the Global Climate (ERA5) was utilized. Copernicus Climate Change Service (C3S) at ECMW produces ERA5 data. From 1979 to the present, ERA5 climatic data are available on a 0.25° × 0.25° grid. The monthly data for precipitation and temperature covering the period from 2003 to 2018 were obtained from https://cds.climate.copernicus.eu/, accessed on 14 October 2020.

### 2.3. Methodology

In our study, MODIS 8-day data products were used, specifically the Combined Terra-Aqua (MOYDGL06*) dataset. This is created through the combination and merging of 8-day snow cover product (version 6) Terra-MOD10A2 and Aqua-MYD10A2 derived MODIS sensor images with RGI6.0. It has a spatial resolution of 500 m and a temporal resolution of 8 days. The nearest neighbouring resample method is used to level all the data into coarse resolution (500 m) for pixel-by-pixel comparison. Figure 2 shows how the proposed study removes the clouds as part of its methodology which involves various steps and filters as explained by the following: (a) seasonal filtering, (b) temporal filtering, (c) spatial filter, (d) merging Terra- and Aqua filtered to generate snow product, and (e) integrating glaciers (RGI 6.0) in the improved snow product.

#### 2.3.1. Seasonal Filtering

The data were categorized into snow and no snow images, and then further divided into summer and winter seasons for each hydrological year. The data from both seasons were combined, and the maximum range of snow cover was identified for each season. Based on this information, the MOD10A2 and MYD10A2 data were extracted. Pixels affected by clouds beyond the maximum snow extent were adjusted to indicate the absence of snow. This method aimed to enhance efficiency by reducing processing time and addressing uncertainties associated with cloud removal, utilizing temporal and spatial filters.

#### 2.3.2. Temporal Filtering

The cloud elimination process involved a seasonal filter followed by a temporal filter substituting cloudy pixels with clear pixels [45,46,47]. The seasonal filter effectively eliminated a significant portion of the clouds, while the remaining clouds were addressed using the temporal filter. The purpose of the temporal filter was to replace cloudy pixels with clear ones by analysing multiple images. For this task, up to four images (two preceding and two subsequent 8-day composite images) were selected. Each cloudy pixel was compared to its corresponding pixel in the subsequent image to determine the presence of snow or no snow. If the subsequent pixel provided a conclusive indication, the cloudy pixel was replaced accordingly. In cases where the subsequent image did not offer a definitive conclusion, the pixel in the previous image was examined using similar criteria. This process was repeated for up to two preceding and two subsequent images in cases of persistent cloud cover. If clouds persisted across all four images, the spatial filter was employed to eliminate the remaining cloudy pixels. The temporal filter operated under the assumption that snow cover remained constant even in continuous cloudy conditions, disregarding minimal melting that might occur. Sequential Equations (1)–(3) outlined the steps of the temporal filter, enabling the conversion of cloudy pixels to no snow if subsequent equations identified the pixels as no snow [45]. Equation (1) required replacing of the “OR” operator with “AND” to satisfy the condition for snow to no snow. Conversely, if Equation (3) indicated the presence of clouds in all conditions, the pixels remained cloudy within the temporal filter and were then considered for conversion to snow or no snow using the spatial filter.

Step 1:(1)$$Z_{\left(a,b,r\right)}^{d}=snow\;IF\;\left(\left.Z_{\left(a,b,r-1\right)}=snow\;OR\;\right.\;\left.Z_{\left(a,b,r+1\right)}=snow\right.\right)$$

Step 2:(2)$$Z_{\left(a,b,r\right)}^{d}=snow\;IF\;(\left.Z_{\left(a,b,r-1\right)}\;AND\right.\;\left.Z_{\left(a,b,r+1\right)}=Cloud\;\;\right.\left.\;AND\;\right.Z_{\left(a,b,r-2\right)}=snow)$$

Step 3:(3)$$Z_{\left(a,b,r\right)}^{d}=snow\;IF\;(\left.Z_{\left(a,b,r-1\right)}\;AND\right.\;\left.Z_{\left(a,b,r+1\right)}\;\;AND\;\right.Z_{\left(a,b,r-2\right)}=cloud\;\;AND\;Z_{\left(a,b,r+2\right)}=snow)$$

In the mathematical expressions above, the matrix is represented by Z, with d indicating clouds, and b and a representing the row and column indices of Z. Additionally, r denotes the time index.

#### 2.3.3. Spatial Filtering

After applying the temporal filter to the images, the remaining cloudy pixels underwent the majority neighbourhood spatial filter. The reason for applying the spatial filter after the temporal filter is its effectiveness in eliminating small or fragmented clouds. By examining the majority classification of the neighbouring non-cloudy pixels, the cloudy pixel is reclassified as either snow or no snow [45]. In cases where there is an equal number of no snow and snow pixels surrounding a particular pixel, that pixel is designated as snow. Furthermore, running this filter once does not eliminate all the remaining cloudy pixels, so the filter was applied three times. Cloudy pixels persist only when all eight neighbouring pixels are also cloudy.

#### 2.3.4. Combining Snow Products (Filtered Terra and Aqua Images)

Then, the integration of Terra and Aqua datasets was carried out with a focus on accurately identifying snow-covered pixels. This involved considering only the pixels classified as snow in both datasets and cross-verifying the snow mapping between them. By adopting this criterion, the researchers were able to mitigate uncertainties that could arise from cloud removal techniques employed in either the Terra or Aqua data. This step significantly improved the quality of the snow product. Consequently, cloud cover was effectively eliminated from all the analysed images throughout the study period using the described methodology. The procedure for merging snow data obtained from Terra and Aqua satellites is detailed in Equations (4) and (5) [41].

Step 1:(4)$$Z_{\left(a,b,r\right)}^{combined}=snow\;IF\;\{\left(Z_{\left(a,b,r\right)}^{Y}=snow\;OR\right)\;cloud\;AND\;\left(Z_{\left(a,b,r\right)}^{X}=snow\right)\}$$

Step 2:(5)$$Z_{\left(a,b,r\right)}^{combined}=snow\;IF\;\{\left(Z_{\left(a,b,r\right)}^{Y}=snow\right)\;\;AND\;\left(Z_{\left(a,b,r\right)}^{X}=snow\;OR\;Cloud\right)\}$$

In the provided context Y refers to the final product obtained from the Terra satellite while X represents the final products obtained from the Aqua satellite.

#### 2.3.5. Combined Glaciers (RGI 6.0) in the Improved Snow Product

In areas where snow and glaciers coexist, accurately differentiating between them is challenging, especially during the accumulation period. The widely used MODIS algorithm for snow detection faces difficulties in identifying glacier ice as it melts and the glacier surface has a low albedo. Additionally, the algorithm struggles to map ice covered by debris. To overcome these challenges, ref. [44] incorporated the latest version of the RGI 6.0 and supraglacial debris cover data into the MODIS dataset. These datasets were adjusted to match the pixel size of MODIS and combined to create a comprehensive product that encompassed both snow and glacier cover, including debris-covered and debris-free areas. This product is particularly valuable for applications related to glacio-hydrology.

The methodology for estimating snow cover area (SCA) is depicted in Figure 2. The study used cloud-free satellite data from the Terra satellite’s Moderate Resolution Imaging Spectroradiometer (MODIS) sensor (version 6) combined with Terra-Aqua MODIS (MOYDGL06*) from 2003 to 2018. The data was then processed using Erdas Imagine 2015 software, which transformed the data items into Universal Traverse Mercator (UTM) projection using the World Geodetic System-1984 (WGS84) datum. The area of interest was then clipped from these images. An improved final snow product was developed to approximate the SCA; the product included flags indicating transitions between snow and non-snow states within each pixel. There were six categories of flag values: 0 represented non-snow pixels, 200 represented pixels with consistent snow presence, −200 represented pixels transitioning from snow to non-snow, 210 represented pixels transitioning to snow due to cloud cover, 240 represented debris-covered pixels, 250 represented debris-free ice pixels. Snow pixels were isolated and extracted for mapping and snow cover analysis. A total of 736 cloud-free images were obtained and used to indicate SCA. The resulting SCA maps were analysed to observe yearly fluctuations. To classify SCA into different zones, a pixel-level comparison was conducted with elevation, slope, and aspect maps derived from ASTER GDEM. The SCA map was compared pixel-by-pixel with the elevation, slope, and aspect maps to categorize SCA into distinct zones.

### 2.4. Statistical Analysis

Statistical analysis was conducted using the Mann–Kendall test [48] to find the significance between the SCA and the climatic parameters. The Mann–Kendall test is a non-parametric statistical test that is employed to determine the presence or absence of a trend in a time series data. A significant *p*-value indicates that there is evidence of a trend in the data, while a non-significant *p*-value suggests that there is no significant trend. The Mann–Kendall test can also be used to identify the direction of the trend, i.e., whether it is increasing or decreasing over time. Using Equation (6) the MK test statistics $\left(y_{s}\right)$ are determined [49]:(6)$$\mathrm{if}\;\left|y\right|>y_{\alpha ∕2}$$

If we set the significance level (α) to be 5%, then the critical value for a two-tailed test is 1.96, which corresponds to a *p*-value of 0.025. For a calculated test statistic of greater than 1.96 or less than −1.96, we reject the null hypothesis and conclude that there is a significant trend in the data.

Similarly, Sen’s slope [50] is a non-parametric method for estimating the slope or trend of a univariate time series by using Equation (7):(7)$$S_{i}=median\left(\frac{Y_{a}-Y_{b}}{v_{a}-v_{b}}\right)\;$$

Here the variables $Y_{a}$ and $Y_{b}$ represent the data values at time points $v_{a}$ and $v_{b}$, respectively. The values of Sen’s slope for a specific data point, denoted as $S_{i}$, are calculated for each i ranging from 1 to Z. A positive value of Sen’s slope indicates a rising trend in the data, while a negative value indicates a decreasing trend. These trend assessments are made with a confidence level of 95%.

## 3. Results and Discussion

Using the MODIS snow product, SCA variation on a geographical and temporal basis was calculated for the Beas River basin from 2003 to 2018. The primary objective of this study was to determine the changes in annual and seasonal SCA in the Beas River basin by utilizing MODIS 8-day data products, specifically the Combined Terra-Aqua (MOYDGL06*) dataset. The present study aimed to investigate the correlation between SCA, temperature, and precipitation in the Beas River basin. The key study findings are presented and discussed in the following sections.

### 3.1. Variation of SCA Annually

The SCA for each month was determined by averaging the eight-day snow products received during that specific month. Figure 3 presents the SCA map specifically for the year 2014. The data reveals that the average SCA for the entire period from 2003 to 2018 was 3017.3 km^2^, accounting for 56.03% of the total area. Table 2 shows the computed SCA for the Beas River basin from 2003 to 2018 over twelve months. From Table 2, the highest annual SCA was recorded in 2014 (61.84%), while the lowest annual SCA was recorded in 2016. (49.22%). In comparison to the period from 2003 to 2015, the annual SCA increased at a lower pace during 2016–2018, whereas a sharp reduction occurred between 2015–2018. The yearly SCA likewise showed an upward tendency from 2009 to 2015. Figure 4 shows the annual variation of SCA during 2003–2018 in the Beas River basin.

### 3.2. Seasonal Variation of SCA

The SCA fluctuations exhibited seasonality throughout the year, with distinct patterns during the winter, pre-monsoon, monsoon, and post-monsoon seasons. Specifically, these seasons occur from December to March, April to June, July to September, and October to November, respectively [51,52]. In July, August, and September, SCA was 36.58%, 30.16%, and 33% for the whole basin area, respectively, which continued throughout the year and increased to 79.93% and 84.45% for January and February as shown in Table 2.

Figure 5 illustrates the seasonal changes in SCA within the Beas River basin from 2003 to 2018. The data reveals that during the winter season, the Beas River basin experienced substantial snow coverage, with approximately 82.19% of the area being covered by snow throughout the mentioned time frame. In the pre-monsoon, monsoon, and post-monsoon seasons, the SCA percentages were 62.03%, 33.24%, and 53.41%, respectively. In winter, pre-monsoon season, and post-monsoon season SCA trend are positive whereas in the monsoon season SCA trend is negative.

### 3.3. Relation between Elevation and SCA

The variation of SCA was analysed across different elevation zones from 2003 to 2018, and a specific SCA map for the year 2014 is depicted in Figure 6. Table 3 provides information on the highest, lowest, and average SCA values recorded in various elevation zones. In the fifth elevation range, the maximum average SCA was determined to be 916.28 km^2^. On the other hand, the elevation range of 5442–6582 m a.m.s.l exhibits the lowest average SCA of 72.42 km^2^. The fifth elevation zone demonstrates the highest average SCA, followed by the sixth and fourth elevation zones. The monthly variation of average SCA in the Beas River basin across different elevation zones from 2003–2018 is depicted in Figure 7.

In several years, the SCA in the first and second elevation zones of the Beas River basin was less than 6%, although no elevation zone was snow-free even during the depletion period. The correlation between SCA and various elevation zones was investigated by analysing the average SCA for each month in different elevation zones spanning from 2003 to 2018. Curve fitting was carried out using first-order polynomial and regression coefficients ranging from 0.08 to 0.34. In all months, elevation was highly associated with SCA, according to our analysis. From December to March, the trend in different elevation zones was relatively uniform, but the gradient steepness progressively increased from April to September.

### 3.4. Relation between SCA and Slope

The study also examined how SCA varies across various slope zones to observe changes in its patterns. Figure 8 illustrates the SCA distribution for each slope zone during each month of 2014. Table 4 provides a summary of the maximum, minimum, and average SCA values observed in various slope zones. The third slope zone (30–40°) exhibits the highest SCA of 861.26 km^2^, while the seventh slope zone (60–81°) shows the lowest SCA of 2.12 km^2^. The percentage variation in SCA is consistent across all slope zones during the accumulation period. Furthermore, the reduction in SCA percentage in the third and fourth slope zones is more significant in June, July, August, and September (as depicted in Figure 8). The lower elevation regions contribute to a greater decrease in SCA, known as the ablation zone. Given that 75% of hazardous avalanches occur in the 32–40° slope zone in the mid-latitude Himalayan range and 60% of snow avalanches occur in the lower Himalayan zone in the 30–35° range [53], the slope range of 30–40° (zone 4) holds particular significance. The regression coefficient value is greater than 0.2 in August and September, indicating that the percentage of SCA for zones 1 to 3 initially decreases, then increases until the sixth zone, but subsequently decreases again. The downward trend is attributed to the gravitational sliding of snow over a 60° slope. Month-wise variation of average SCA in km^2^ for the Beas River basin from 2003–2018 in slope zones is depicted in Figure 9.

### 3.5. Relation between Aspect and SCA

The average SCA for each aspect was calculated yearly and is presented in Table 5. Figure 10 displays the SCA distribution for each aspect zone during each month of 2014. It can be observed that most zones have relatively similar SCA values, except for the fifth zone (south direction) which has the highest SCA of 595.19 km^2^, and the second zone (northeast direction) which has the lowest SCA of 235.05 km^2^. The second zone also exhibits the lowest SCA percentagewise, with only 4.37% coverage, followed by the northeast-facing zone. A first-order polynomial fit was used to analyse the relationship between aspect and monthly SCA, with regression coefficients ranging from 0.08 to 0.34 for all months. The most significant reduction in SCA occurred in July, August, and September in the northern and northeastern aspects. Figure 11 shows the month-wise variation of average SCA in km^2^ for the Beas River basin from 2003–2018 in the aspect zones.

### 3.6. Relation between Climate Data and SCA

The ERA5 temperature and precipitation dataset from 2003 to 2018 was analysed using the Mann–Kendall and Sen’s slope tests to assess trends [48,50]. Table 6 provides a summary of the findings for these tests on an annual and seasonal basis. The annual mean temperature and SCA have a negative correlation, indicated by a correlation coefficient of −0.56. This negative correlation is statistically significant at the 0.01 level, suggesting that SCA has decreased over the past 16 years due to rising temperatures. On the other hand, the annual precipitation and SCA exhibit a positive correlation, with a correlation coefficient of 0.35. Although this correlation is statistically weak, it implies a positive relationship between SCA and precipitation from 2003 to 2018. Figure 12 illustrates that in 2016, the highest temperature and lowest precipitation led to the lowest annual SCA during the study period. Conversely, in 2014, the lowest temperature and relatively high precipitation resulted in the highest SCA. Therefore, fluctuations in annual SCA are influenced by both temperature and precipitation. Overall, the relationship between SCA and temperature was stronger during the period from 2003 to 2018 compared to the association between SCA and precipitation.

The primary objective of this study was to examine the variations in annual and snow cover area (SCA) within the Beas River basin using MODIS 8-day data products, specifically the Combined Terra-Aqua (MOYDGL06*) dataset. Over recent years, the study area has undergone noticeable changes, including a decrease in SCA and total precipitation, accompanied by a rise in temperature. When considering the average SCA for the Beas River basin yearly, the highest percentage was recorded in 2014 at 61.84%, while the lowest SCA was observed in 2016 at 49.22%. Throughout the entire period from 2003 to 2018, the average SCA across the years amounted to 55.37% of the total basin area. Focusing on individual months, February exhibited the highest average SCA of 84.45%, whereas August had the lowest SCA of 30.16% during the same period (2003–2018).

The Pearson correlation analysis demonstrated a robust connection between snow cover area (SCA) and climatic parameters. As temperatures increased, SCA decreased, whereas higher precipitation led to an expansion in SCA. The annual assessment revealed the varying influence of these climatic factors, with temperature exhibiting a stronger association with SCA compared to precipitation. The correlation between temperature and SCA was more pronounced than that between precipitation and SCA. Figure 13 shows the monthly precipitation for the Beas River basin for the years 2003–2018. Particularly during the monsoon and post-monsoon seasons, a highly significant correlation was observed between SCA and climatic parameters. Additionally, the study suggests that in situations where on-site data is unavailable, the ERA5 dataset can serve as a reliable alternative for trend analysis and correlation studies [12].

## 4. Conclusions

The present research focused on examining the variations in seasonal snow cover in the Himalayan region between 2003 to 2018, taking into account different elevations, slopes, and aspects. The study findings suggest that the climate change caused by global warming has directly or indirectly affected the regional climate of the Himalayas, particularly in the lower and middle Himalayan zones [37]. This has led to changes in the timing of snowmelt runoff and snow accumulation periods. These changes are attributed to the rising temperatures in the Himalayan Mountain range. The primary objective of this study was to explore the relationship between the SCA, temperature, and precipitation in the Beas River basin. Recent years have shown an upward trend in the study area, characterized by a decrease in SCA and total precipitation, along with an increase in temperature. These trends indicate significant climate changes that may have adverse effects. The study examined the fluctuating patterns of snow coverage throughout different seasons, offering valuable knowledge for tasks such as mapping and assessing snow avalanches, managing hydrology, and handling water resources. Furthermore, it also offers valuable directions for accurately predicting natural hazards and safeguarding water resources. Overall, the future of finding snow cover areas involves leveraging technological advancements, incorporating data into models, and fostering interdisciplinary collaboration to enhance our understanding of snow dynamics and their implications for various sectors, including hydrology, water resources management, and climate change research.

## Figures and Tables

**Figure 1 sensors-23-08387-f001:**
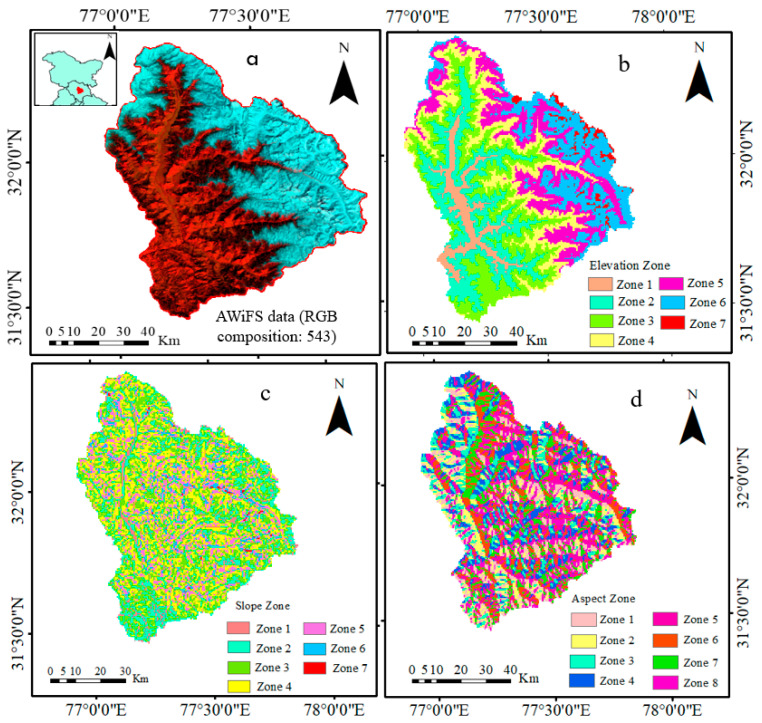
This depicts the (**a**) study area (Beas Basin), (**b**) elevation zones, (**c**) slope zone, and (**d**) aspect zone.

**Figure 2 sensors-23-08387-f002:**
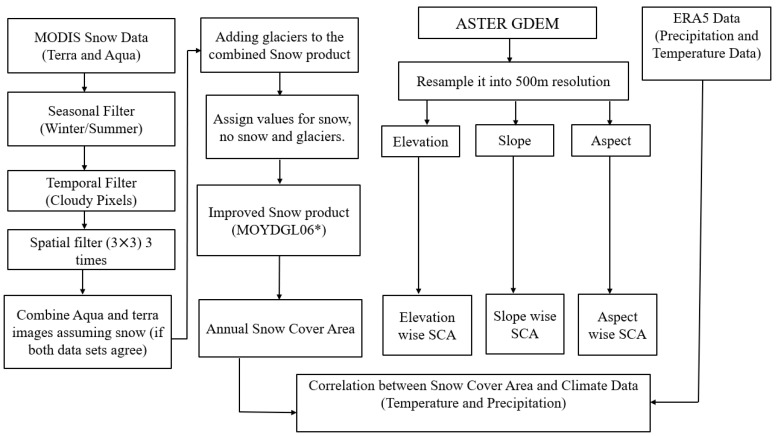
Flowchart of methodology to analyse the snow cover change with respect to different parameters.

**Figure 3 sensors-23-08387-f003:**
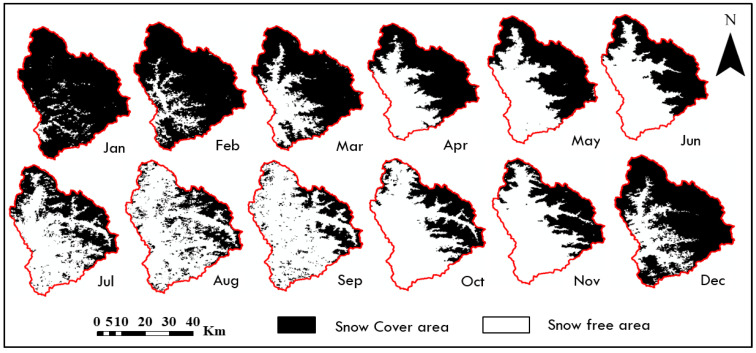
Monthly mean SCA for the year 2014 over Beas River basin.

**Figure 4 sensors-23-08387-f004:**
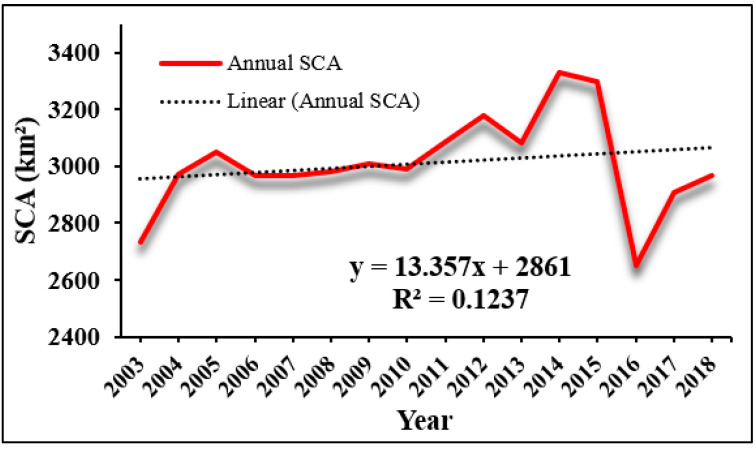
Annual variation of SCA during 2003–2018 in Beas River basin.

**Figure 5 sensors-23-08387-f005:**
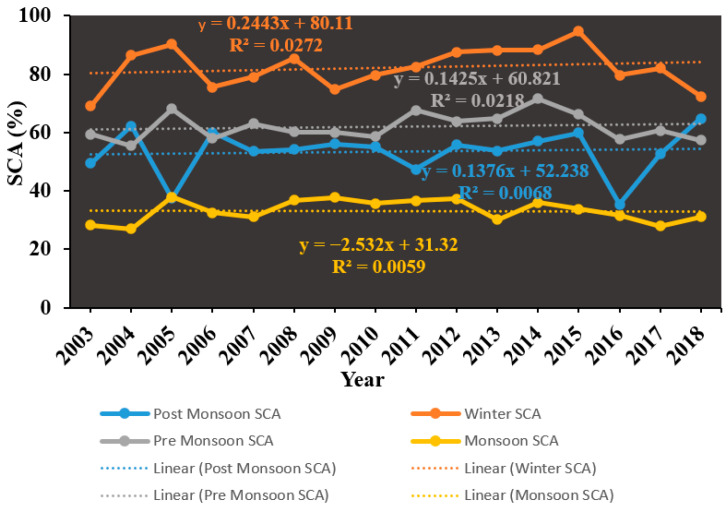
Seasonal variation of SCA in Beas River basin between 2003 and 2018.

**Figure 6 sensors-23-08387-f006:**
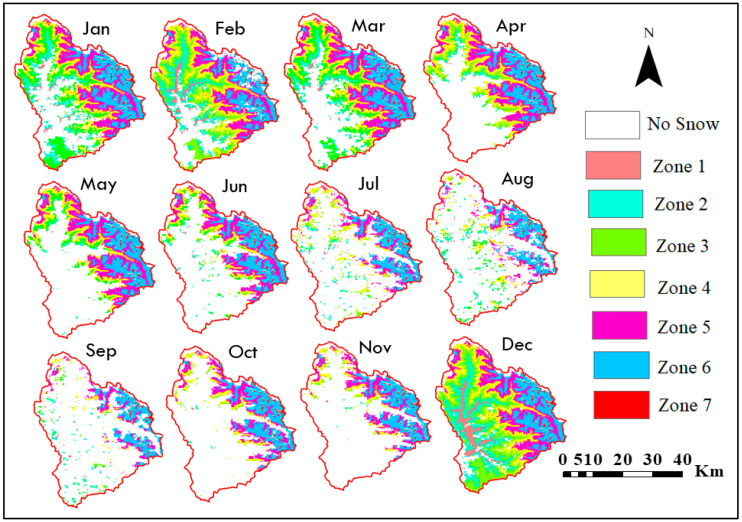
Maps depicting SCA categorized by elevation in the Beas River basin for the year 2014.

**Figure 7 sensors-23-08387-f007:**
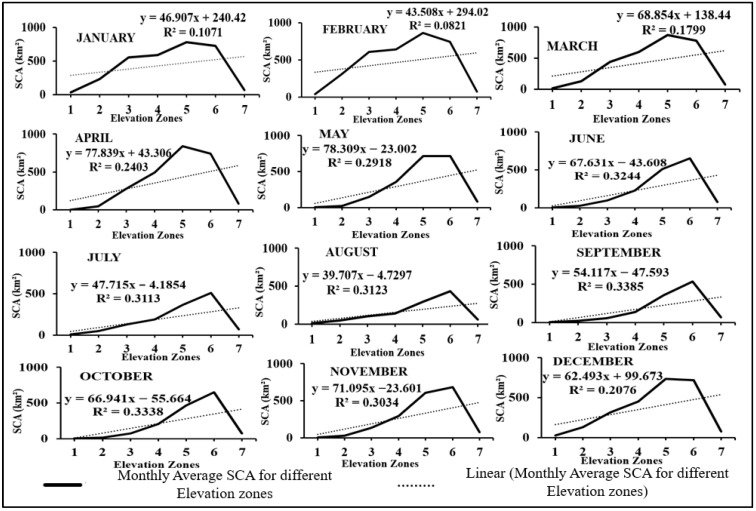
Monthly variation of average SCA in km^2^ for Beas River basin from 2003–2018 in elevation zones.

**Figure 8 sensors-23-08387-f008:**
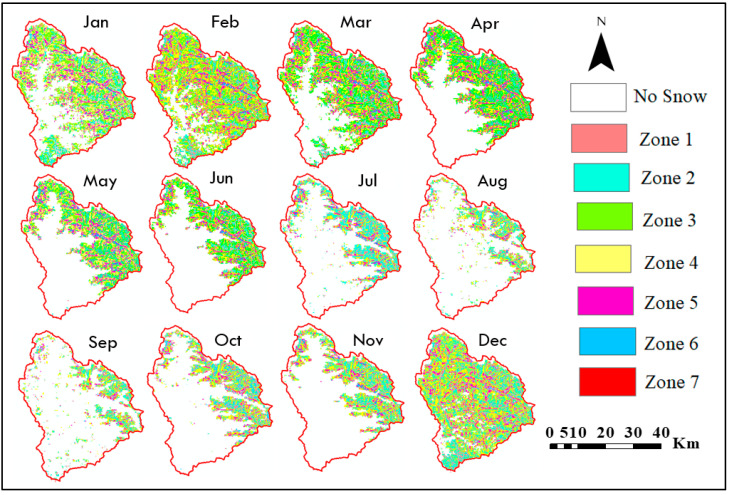
Slope-wise SCA maps for 2014 for the Beas River basin.

**Figure 9 sensors-23-08387-f009:**
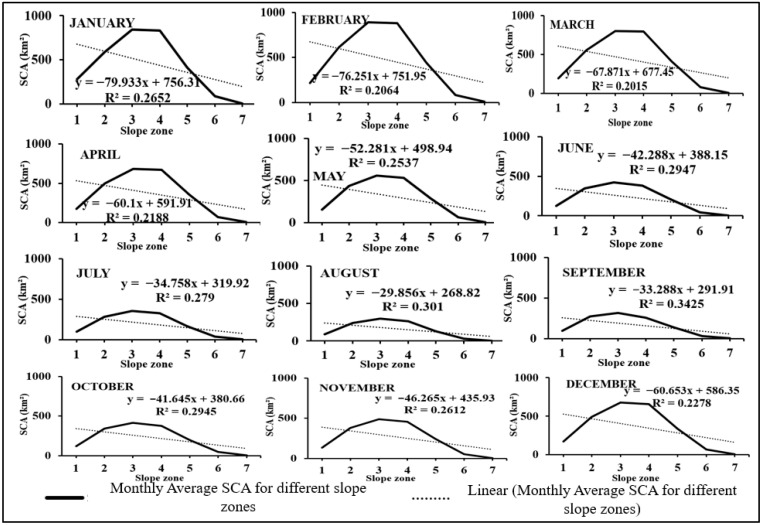
Month-wise variation of average SCA in km^2^ for Beas River basin from 2003–2018 in slope zones.

**Figure 10 sensors-23-08387-f010:**
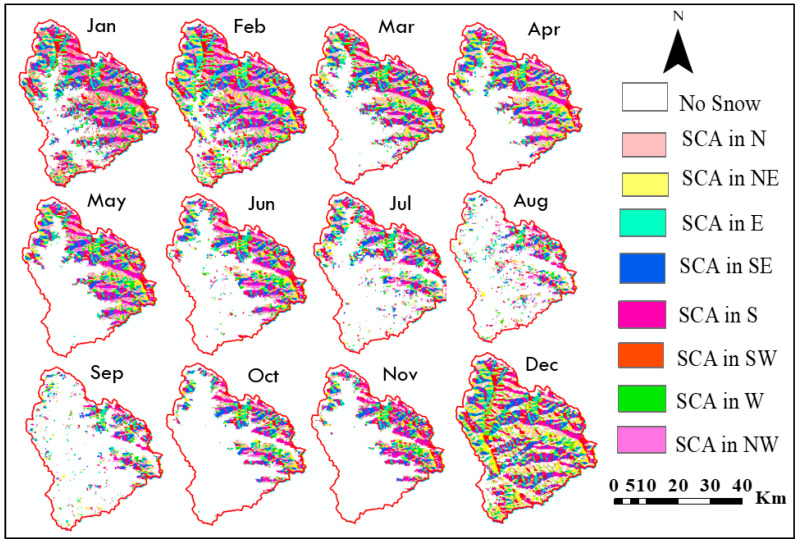
Aspect-wise SCA maps for 2014 for the Beas River basin.

**Figure 11 sensors-23-08387-f011:**
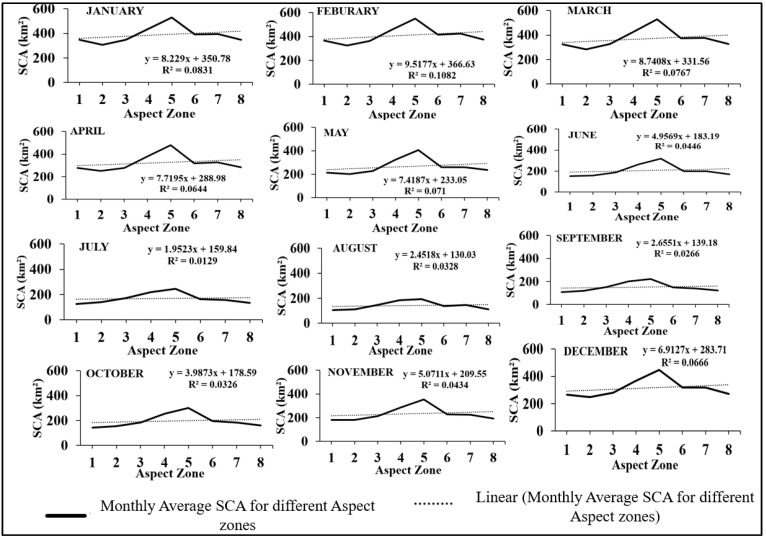
Month-wise variation of average SCA in km^2^ for Beas River basin from 2003–2018 in aspect zones.

**Figure 12 sensors-23-08387-f012:**
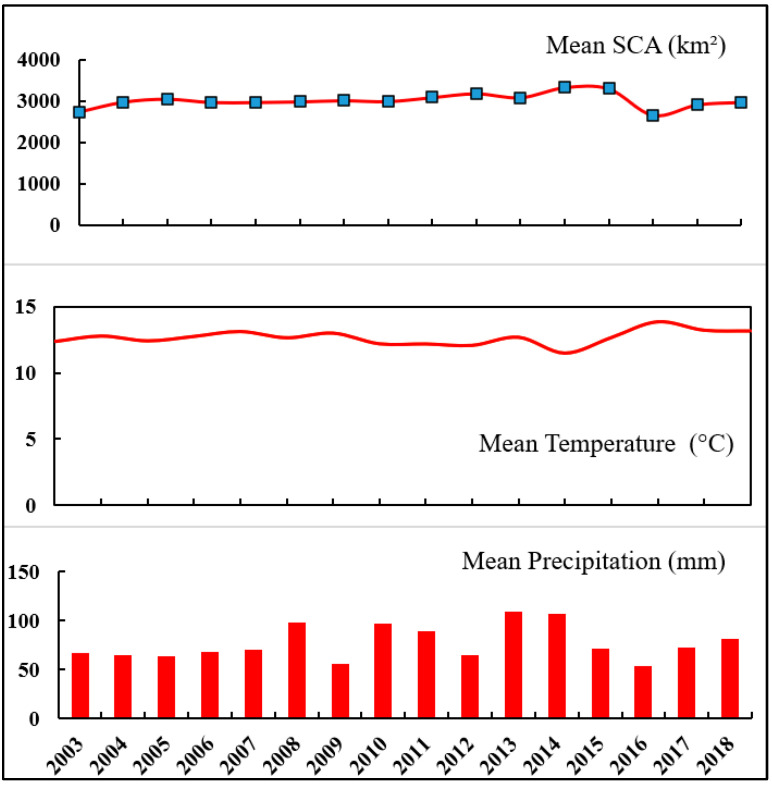
Annual relation between SCA and temperature and precipitation.

**Figure 13 sensors-23-08387-f013:**
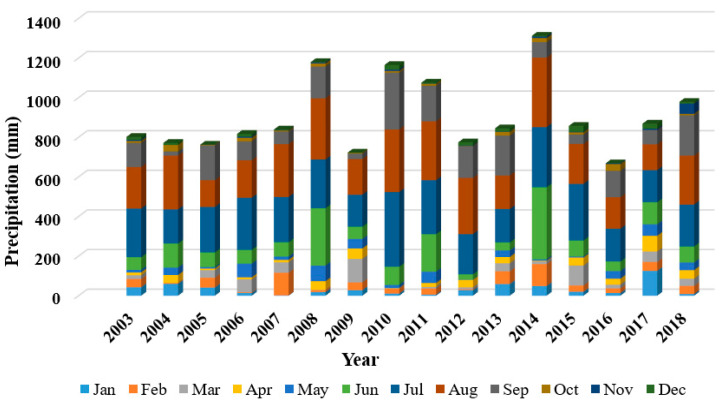
Monthly precipitation for the year 2003–2018.

**Table 1 sensors-23-08387-t001:** (**a**) Details of elevation zones of Beas River basin. (**b**) Details of slope zones of Beas River basin. (**c**) Details of aspect zones of Beas River basin.

**(a)**		
**Zone of Elevation**	**Elevation Range (m)**	**Area of Zone (km^2^) (%)**
1	853–1618	414.21 (7.69%)
2	1618–2383	1149.10 (24.34%)
3	2383–3148	1219.34 (22.64%)
4	3148–3912	908.54 (16.87%)
5	3912–4677	1107.57 (20.57%)
6	4677–5442	571.47 (10.64%)
7	5442–6582	13.46 (0.25%)
Total		5384 (100%)
**(b)**		
**Zone of Slope**	**Slope Range (°)**	**Area of Zone (km^2^) (%)**
1	00–10	506.23 (9.40%)
2	10–20	1192.25 (22.14%)
3	20–30	1944.85 (36.15%)
4	30–40	1269.96 (23.58%)
5	40–50	414 (7.69%)
6	50–60	53.86 (1%)
7	60–80	2.32 (0.04%)
Total		5384 (100%)
**(c)**		
**Zone of Aspect**	**Aspect**	**Area of Zone (km^2^) (%)**
1	N	688.19 (12.78%)
2	NE	604.27 (11.22%)
3	E	610.31 (11.33%)
4	SE	670.31 (12.45%)
5	S	726.12 (13.48%)
6	SW	692.21 (12.85%)
7	W	705.84 (13.14%)
8	NW	686.45 (12.75%)
Total		5384 (100%)

**Table 2 sensors-23-08387-t002:** Annual SCA (%) in Beas River basin during 2003–2018.

Month	2003	2004	2005	2006	2007	2008	2009	2010	2011	2012	2013	2014	2015	2016	2017	2018	Average
January	49.39	85.3	87.74	77.97	72.97	77.46	74.52	73.01	93.98	87.24	82.85	89.71	95.56	76.26	85.38	69.49	79.93
February	88.8	87.69	92.8	72.94	85.12	93.03	74.88	86.19	71.1	87.78	93.5	86.93	93.75	82.89	78.71	75.09	84.45
March	78.6	68.12	85.63	78.74	91.01	75.77	68.78	73.7	84.09	81.76	83.02	85.82	75.47	76.64	79.72	72.85	78.73
April	71.63	65.96	72.04	65.89	69.61	69.21	67.53	59.92	74.52	66	67.5	78.45	75.81	64.22	64.99	65.32	68.66
May	52.65	50.31	61.45	51.05	52.92	56.44	57.04	52.36	61.63	62.03	60.08	67.59	62.72	51.18	52.86	55.32	56.73
June	34.85	37.45	53.45	36.43	38.7	39.49	46.67	48.37	50.37	45.2	48.47	54.37	50.99	38.35	44.86	36.07	44.06
July	26.4	26.75	46.74	36.37	30.66	33.47	35.52	34.84	41.04	40.22	36.6	47.81	39.58	39.71	34.73	34.84	36.58
August	29.73	22.95	35.11	29.1	32.94	32.13	32.87	38.42	38.52	33.92	26.42	30.22	30.07	23.29	21.98	24.97	30.165
September	28.52	31.15	32.07	32.25	29.77	44.94	44.87	33.97	30.26	37.6	27.64	30.24	31.69	31.84	27.05	33.54	32.96
October	28.34	67.35	36.72	44	47.05	50.8	38.27	50.03	38.64	42.05	36.26	44.4	44.52	25.08	38.03	49.73	42.58
November	45.32	54.58	36.1	56.69	40.4	51.71	63.5	55.65	42.68	49.36	58.06	44.95	60.98	22.38	50.55	73.66	50.41
December	74.53	64.87	39.49	79.59	73.25	59.98	66.52	59.51	60.86	75.66	66.9	81.6	74.18	58.85	69.12	70.97	67.24
Annual SCA (%)	50.73	55.21	56.61	55.08	55.38	57.04	55.91	55.50	57.31	59.07	57.27	61.84	61.28	49.22	53.99	55.15	56.04
Annual SCA (km²)	2731	2973	3048	2966	2981	3071	3010	2988	3086	3180	3083	3329	3299	2650	2907	2969	3017

**Table 3 sensors-23-08387-t003:** The average SCA for different elevation zones in the Beas River basin from 2003 to 2018.

Elevation Zone	Elevation Range (m)	Mid Elevation (m)	SCA (Min) (km^2^)	SCA (Max) (km^2^)	Average SCA (km^2^)
1	853–1618	1235.5	2.35	38.46	119.88
2	1618–2383	2000.5	13.77	312.31	224.69
3	2383–3148	2765.5	57.88	592.32	553.95
4	3148–3912	3530	135.96	606.11	569.41
5	3912–4677	4294.5	290.03	872.34	916.28
6	4677–5442	5059.5	434.34	589.29	560.67
7	5442–6582	6012	59.29	78.89	72.42
Total					3017.3 km^2^ (56.03%)

**Table 4 sensors-23-08387-t004:** Slope zone-wise SCA during 2003–2018.

Slope Zone	SCA (Min) (km^2^)	SCA (Max) (km^2^)	Average SCA (km^2^)
1	85.72	278.51	234.09
2	240.13	713.76	657.23
3	294.2	889.03	861.26
4	263.96	898.99	835.73
5	130.78	490.05	375.96
6	28	54.9	50.91
7	2.96	3.82	2.12
			3017.3 km^2^ (56.03%)

**Table 5 sensors-23-08387-t005:** Aspect zone-wise SCA during 2003–2018.

Aspect Zone	SCA (Min) (km^2^)	SCA (Max) (km^2^)	Average SCA (km^2^)
1	271.29	349.13	304.29
2	269.85	305.96	235.05
3	302.12	368.63	349.59
4	376.67	447.83	427.12
5	441.16	627.45	595.19
6	305.65	399.80	376.12
7	315.76	401.41	394.99
8	178.62	373.74	334.95
			3017.3 km^2^ (56.03%)

**Table 6 sensors-23-08387-t006:** Temperature and precipitation trend analysis.

Season	Precipitation Z	Q (°C/Year)	Trend (95%)	Temperature Z	Q (°C/Year)	Trend (95%)
Winter	−1.10 *	−0.965	Falling	0.00	0.063	No Trend
Pre-monsoon	0.267 *	1.746	Rising	0.033 *	0.050	Rising
Monsoon	0.167 *	1.313	Rising	0.517 *	0.066	Risings
Post-monsoon	0.183 *	0.341	Rising	0.150 *	0.048	Rising
Annual	0.233 *	1.892	Rising	0.117 *	0.029	Rising

Q: Sen’s slope, Z: Mann–Kendall test statistics. * Insignificant Trend.

## Data Availability

The MODIS improved snow cover product (MOYDGL06*) was obtained from the website https://doi.pangaea.de/10.1594/PANGAEA.901821 (accessed on 11 November 2020), the ERA5 data were obtained from https://cds.climate.copernicus.eu/ (accessed on 14 October 2020) and DEM was obtained from (https://earthexplorer.usgs.gov/, accessed on 24 January 2021).

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
