# Peer review of "Snow Cover Response to Climatological Factors at the Beas River Basin of W. Himalayas from MODIS and ERA5 Datasets"

_sensors, 2023, doi:10.3390/s23208387_

Round 1

Reviewer 1 Report

There are some syntax errors in this manuscript. The English of the manuscript should be improved. 

Author Response

Dear Learned Reviewer, 

I am very thankful to you for providing such nice suggestions in order to improve the quality of my research article. Further, I need your favour for the publication of this research article with the improvement by your suggestions. I have amended all your suggestions in the revised manuscript with my best knowledge. If anything left, please let me know. We would surely incorporate in the next revision of this manuscript. 

Regards

Dileep

Reviewer 2 Report

This paper “Snow cover response to climatological factors at the Beas River basin of W. Himalayas from MODIS and ERA5 datasets” analyzed the snow cover area in the Beas River basin in the Western Himalayas from 2003 to 2018. We also examine the sensitivity of snow cover to temperature and precipitation. The authors used MODIS-based MOYDGL06 products for snow cover estimation and the European Centre for Medium-Range Weather Forecasts (ECMWF) Atmospheric Reanalysis of the global climate (ERA5) for climate data. Their results showed a strong correlation between temperature and snow cover area, indicating that temperature plays a significant role in snow dynamics. These findings have important implications for water resource management, climate studies, and disaster management. The application of this paper is interesting, but those flaws should be fixed before a proper review. My comments are as follows.

The structure of Introduction can review more recently published papers, especially novel applications they have provided in snow cover product generation based on remote sensing. For example, “remote sensing” journal provided and reviewed a lot of similar articles in the field of snow cover responses to climate change.

The accuracy of precipitation data is important, and different precipitation products have their own advantages. Authors can evaluate the CHIRPS product with other precipitation products, such as MSWEP, TMPA or GPCP products, to demonstrate the necessity of using the CHIRPS product instead of other precipitation products.

Authors can also outlook the producing of IDF curves with the fusion of multiple remote sensing data with novel technologies, such as deep learning method. Most of the fused precipitation product outperform single precipitation product.

Line 208: Equation (2) should be placed appropriately.

Line 263: in Figure 2, the flowchart of methodology is ambiguous, and some arrow symbols should be placed logically. For example, the arrow symbol between “annual snow cover area” and “correlation between snow cover area and climate data” is not correct.

Line 274: Equation (6) should add the results after the condition statement or change the following text into equations.

Line 306: In figure 3, boundary of the study area should be added to distinguish snow free area and blank space. The figures below, such as figure 6,8,10, have the same issue.

Line 327: In table 2, “*Average annual SCA for 2003-2018 is 3017.30 sq. km (~56%)” is not necessary.

Line 427: In my opinion, Figure 12 (a) and (b) can be merged into one figure to illustrate the relation between temperature and precipitation.

Minor editing of English language required.

Author Response

(The authors gave the same response as above.)

Round 2

Reviewer 1 Report

Most of the mistakes had been corrected in this manuscript and the additional information added to manuscript made the text more readable.

To make the manuscript publishable, further minor modifications are required.

1.       Lines 145-146: The legend for the color of elevation is still missing in Figure 1 (a).

2.       If there is no direct correlation has been done between figure 7, figure 9, and figure 11, why there are trend lines in these figures?

The quality of English is acceptable.

Author Response

I am very thankful to the learned reviewer for dedicating their valuable time to review my paper and providing constructive comments to enhance the manuscript's quality. Indeed, the quality of this manuscript has improved significantly. I have done my utmost to incorporate your suggestions into the revised manuscript. If there are any further recommendations, please do not hesitate to let me know, and I will certainly include them in the final version of this manuscript. Once again, thank you very much.
